# What's in a Query: Polarity-aware Distribution-based Fair Ranking

## Abstract

Machine learning-driven rankings, where individuals (or items) are ranked in response to a query, mediate search exposure or *attention* in a variety of safety-critical settings. Thus, it is important to ensure that such rankings are fair. Under the goal of equal opportunity, attention allocated to an individual on a ranking interface should be proportional to their relevance across search queries. In this work, we examine amortized fair ranking – where relevance and attention are cumulated over a sequence of user queries to make fair ranking more feasible. Unlike prior methods that operate on expected amortized attention for each individual, we define new divergence-based measures for attention distribution-aware fairness in ranking (DistFaiR), characterizing unfairness as the divergence between the distribution of attention and relevance corresponding to an individual over time. This allows us to propose new definitions of unfairness, which are more reliable at test time and outperform prior fair ranking baselines. Second, we prove that group fairness is upper-bounded by individual fairness under this definition for a useful sub-class of divergence measures, and experimentally show that maximizing individual fairness through an integer linear programming-based optimization is often beneficial to group fairness. Lastly, we find that prior research in amortized fair ranking ignores critical information about queries, potentially leading to a *fairwashing* risk in practice by making rankings appear more fair than they actually are.

## Keywords

fairness, distribution-based fair ranking, query polarity

### ACM Reference Format:

Anonymous Author(s). 2018. What's in a Query: Polarity-aware Distribution-based Fair Ranking. In *Proceedings of Make sure to enter the correct conference title from your rights confirmation emai (Conference acronym 'XX)*. ACM, New York, NY, USA, 14 pages. https://doi.org/XXXXXXX.XXXXXXX

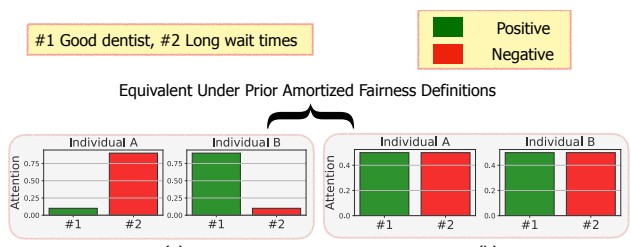

**Figure 1: Past work in amortized fair ranking minimizes the differences between an individual's expected cumulative attention and relevance over a sequence of queries, where queries are considered exchangeable. Critical information about the distributions of attention and properties of queries (such as query polarity) are missing in such a formulation. Our approach, DistFaiR, aims to overcome this. The example shown here considers two search queries with opposite polarities, and a setup where both individuals are equally relevant, but have different attention distributions.**

## 1 Introduction

Automated ranking systems are widely employed in several high-impact settings, such as ordering job candidates, guiding health-related decisions, and influencing purchasing decisions for safety-critical products [10, 17, 52, 59]. These systems directly influence access to critical resources, such as employment opportunities, healthcare, and safe consumer products, all of which significantly affect health and economic outcomes [25, 46], among others. However, prior work has shown that some automated rankings may be biased against some minoritized groups of individuals [8, 26]: for example, women are less likely to occupy higher positions in rankings corresponding to searches made in some online hiring contexts [15]. The increased adoption of large language models (LLMs) as efficient and performant text rankers [24, 30, 47, 63] has the potential to increase the prevalence of automated rankings. However, this expanded usage risks amplifying existing biases in the distribution of user attention and economic opportunity. Mitigating such risks is a critical step towards building a responsible web-based system such as search engines whose performative power to amplify bias has been demonstrated [39].

One domain where automated ranking systems are ubiquitous is *search* [4]. Previous works have proposed several interventions and metrics [7, 21, 41, 62] to ensure that user attention during search is fairly distributed. In these frameworks, ranking algorithms are considered to be mediators of *exposure* to searchers [33, 48], where exposure is defined as the likelihood of *visual attention* from searchers. A common intervention to achieve fair ranking is distributing rankings, and thereby attention, as a function of *relevance* [8, 49].

Notably, fair exposure is impossible in a single ranking where attention decays quicker than relevance, for instance, when all individuals have equal relevance and rankings have position bias w.r.t. attention (see Section 3). As a result, many proposed fairness interventions primarily focus on achieving fair exposure on the aggregate, i.e., over a sequence of queries (e.g., "good dentist", "good optometrists", ...) [8, 49]. Additionally, fairness is *amortized* over a sequence, e.g., query #1 and #2 in Figure 1.

We identify two key limitations in current definitions of fair attention-based amortized ranking: (1) existing methods primarily focus on differences between the mean of attention and relevance distributions across queries, which fails to account for discrepancies in higher-order moments, such as variance or skewness, that may impact fairness (see Figure 1 and Figure 3 in Appendix for intuition), and (2) these methods assume that all attention is inherently positive, overlooking cases where unfairness arises due to disproportionate attention for negative or harmful queries compared to equally relevant counterparts, e.g., Figure 4 in Appendix).

Our approach, *distribution-aware fairness in ranking* (DistFaiR), overcomes these limitations. Our contributions are as follows:

- We formalize a definition of amortized ranking fairness that accounts for differences (beyond means) in the distributions of cumulative relevance and cumulative exposure for individuals over a sequence of queries.
- We identify a set of measures that enable attention and relevance distribution-aware fairness in ranking (DistFaiR). We also consider a worst-case definition of fairness. Specifically, we demonstrate theoretically and empirically that, for these measures, individual fairness upper bounds group fairness for the identified set of DistFaiR measures. Also, we show empirically that individual and group fairness are not at odds, i.e., improving individual fairness often improves group fairness.
- We demonstrate *fairwashing*, a phenomenon where a ranking appears to be more fair than it is when the polarities of queries are not accounted for. We propose polarity-dependent modifications to our newly proposed and existing fairness metrics address this issue of fairwashing.

## 2 Background and Related Work

**Fair Ranking Metrics and Interventions.** Rankings with high ranking quality may be unfair at a group or individual level [8, 12, 16, 19, 40, 49, 60]. Previous works have proposed various techniques to quantify and mitigate unfairness by allocating exposure or visual attention proportionally to relevance at a group or individual level [8, 12, 16, 19, 29, 40, 49, 57, 60]. Some of these are in-processing (i.e., during ranking generation ) [12, 49, 50, 56], while some are post-processing [8] interventions. In some cases, relevance scores used to produce the ranking are jointly estimated along with fairness optimization [12, 40, 50]. Extensive work has also focused on proportion-based ranking, instead of exposure-based ranking [26, 28]. We direct the interested reader to [44] for a detailed review of fairness metrics. While prior papers have proposed some distribution-based measures [19, 25], these have been for fairness under stochastic ranking policies for a single query [27, 28, 49].

In contrast, we focus on the multi-query amortized setup, and consider distributions driven by attention over a sequence of queries, and not (only) due to the stochastic nature of rankings.

**Trade-offs between Group and Individual Fairness.** Prior work has proposed algorithms to optimize individual fairness without violating group fairness or other constraints such as item diversity [23, 25, 27, 27, 28, 28, 46]. Bower *et al.* [12] showed empirically that improving individual fairness is beneficial to improving group fairness in in-processing fair ranking. To the best of our knowledge, no work has theoretically analyzed the relation between group and individual fairness in the amortized setting (e.g., if one bounds the other). In this work, we concretely show that under the proposed definitions of fairness, group unfairness is upper bounded by individual unfairness.

**Impact of Queries in Ranking.** To the best of our knowledge, no prior fairness metrics or interventions utilize information about the query itself in measuring fairness. Closest to our finding is recent work by Patro *et al.* [42], where the authors observe that "user attention may not directly translate to provider utility due to missing context-specific factors" [42]. We expand on this observation, and empirically show that attention-based metrics may fail specifically in a *cross-query amortization* setup. Our finding is also a generalization of a recent finding [18], where it was shown that search results can be manipulated in an amortized setting. Our findings also broadly highlight the risk of fairwashing – maybe due to search engine manipulation – when not considering query polarity. Lastly, while notions of multisided exposure fairness, group over-exposure, and under-exposure [13, 53, 55] are also related to our problem (where the impact of queries or users are considered in fairness formulation), they still assume that all attention is positive. In our work, we propose a method to integrate real-valued query properties such as sentiment polarity into the fairness definition without making these assumptions.

## 3 Amortized Fair Ranking

Given a query $q$ at time $t$ ($q_t$), we consider a ranking task where the goal is to obtain a relevance score $r_i^t \in \mathbb{R}$ for each individual $i$ and order individuals in decreasing order of relevance to the query [45]. The task typically consists of three components: (i) the "query", (ii) the set of individuals to be ranked, and (iii) the relevance scores. Due to position bias, individuals gain exposure based on their position in the ranking, which directly influences the attention they receive [8, 34]. Under the normative principle of equal opportunity, the objective of exposure-based fair ranking is to assign rankings such that the attention allocated to each individual is proportional to their merit [5, 50, 60, 61]. In practical terms, merit is operationalized as a value proportional to relevance.

The concept of amortized fair ranking in existing literature seeks to find a sequence of ranking assignments that minimize the discrepancy between the average cumulative attention and the average cumulative relevance of individuals (or groups) over time. Said differently, relevance and attention are accumulated over a sequence of queries, and the goal is to ensure fairness over this horizon. In this section, we introduce the notations, definitions, and limitations associated with amortized fair ranking for fair attention allocation.

Furthermore, we introduce our distribution and polarity-aware generalization of amortized fair ranking, which results in a more robust solution to the normative goal of equal opportunity.

### 3.1 Notation

Consider a dataset $\mathcal{D}$ of $n$ individuals. Note that we use the term "individual" here interchangeably with any item or entity being ranked throughout the paper. Each individual $i$ belongs to a group $g \in \mathcal{G}$, where $\mathcal{G}$ represents the set of $G$ possible groups. Let $g_k$ denote the $k^{th}$ group in $\mathcal{G}$ where $k \leq G$ and denote group membership as $i \in g_k$ where $g_k \subset \mathcal{D}$ – note that each individual belongs to exactly one group. Denote $q$ to be a sequence of queries, where queries $q_t$ are submitted at discrete time steps $t \in \mathcal{T}$ and $\mathcal{T} = 1, 2, \ldots, T$. A ranking system accepts each of these $T$ queries independently and returns a distinct ranking of $n$ individuals for each query $q_t \in Q$, where $Q$ denotes the space of all queries.

For each individual $i$ at time $t$, let the binary random variables $X_i^t$ and $Y_i^t$ denote the attention and relevance, respectively. Specifically, $X_i^t \sim Bernoulli(a_i^t)$ denotes whether individual $i$ receives attention at time $t$, and $Y_i^t \sim Bernoulli(r_i^t)$ denotes targets for the attention-distribution based on whether individual $i$ is relevant to $q_t$, the query at time $t$. We assume that $X_i^t$ and $X_j^t$ are independent $\forall t$ when $i \neq j$. That is, under the attention models we study, the likelihood of attention is independent of other individuals being ranked, similar to prior work in fair ranking [8, 32]. We also assume that queries are independent. Crucially, for each time step $t$, the total attention and relevance are constrained such that

$$\sum_{i \in n} a_i^t = 1 \qquad and \qquad \sum_{i \in n} r_i^t = 1,$$

such that attention and relevance for individuals are normalized with respect to $n$ individuals at each time step.

Furthermore, denote the cumulative attention and relevance distributions for individual $i$ over the full sequence of queries (all time steps)

$$X_i = \sum_{t \in \mathcal{T}} X_i^t \qquad and \qquad Y_i = \sum_{t \in \mathcal{T}} Y_i^t,$$

respectively.

THEOREM 3.1. *Let* $X_i^t \sim Bernoulli(p_i^t)$ *and*

$$X_i = \sum_{t \in \mathcal{T}} X_i^t.$$

*Then, for any* $\delta > 0$*, we have the following:*

$$P\left(|X_i - \mathbb{E}[X_i]| \geq \delta \mathbb{E}[X_i]\right) \leq 2 \exp\left(-\frac{\delta^2 \mathbb{E}[X_i]}{2 + \delta}\right).$$

REMARK 3.2. *Note that Theorem 3.1's bound depends solely on the expected value of the cumulative attention (and relevance), not the number of queries observed.*

Theorem 3.1 bounds the likelihood of observing a given deviation $\delta$ from the true cumulative attention for an individual over time $t$. We can apply the same exact bound for cumulative relevance.

In our setup, the ranking quality at each timestep $t$ is evaluated using the Discounted Cumulative Gain (DCG) at rank $K$, denoted as DCG@K [31]. The DCG@K score measures the quality of the top-$K$ ranked individuals based on their relevance, adjusting for the rank position using a logarithmic discount factor:

$$\sum_{k=1}^{K} \frac{r_{\text{rank}(k)}^t}{log_2(k+1)},$$

where rank$(k)$ returns the index of the individual at rank $k$.

### 3.2 Motivation For Amortized Fairness Across Different Queries

This work focuses on a class of attention weights where user attention only depends on their ranking position. We assume that attention is proportional to position and follows a distribution informed by domain knowledge. For example, one such distribution used in several prior works is the log-decaying attention distribution [14, 49]. Under this distribution, at time $t$, if an individual $i$ is at position $j$, their attention score $a_i^t \propto \frac{1}{log j}$.

Individual $i$'s attention $X_i^t$ is distributed as $A_i^t = Bernoulli(a_i^t)$, where $a_i^t$ is normalized. Ideally, the probability of an individual receiving attention is proportional to their relevance score $r_i^t$, where relevance scores are $0 - 1$ normalized across all individuals for a given query. Under a fair ranking, individual $i$'s attention should be similarly distributed as their relevance, i.e., $a_i^t \approx r_i^t$. However, as mentioned above, the rate at which attention decays across positions in a ranking is usually very different from the variation in relevance across individuals. This makes it difficult to match the attention distribution to that of relevance within a single ranking. Thus, it may be impossible to achieve the targeted fair attention distribution within a single deterministic ranking [8, 19].

Alternatively, we compare *cumulative* and *amortized* attention and relevance over time. We also assume a more realistic multi-query setup since search systems typically process many queries over time. That is, we consider *online ranking* where a sequence of queries (with corresponding relevance score per individual) arrive over time. We post-process the ranking corresponding to each query (without knowledge of the future queries) to improve fairness.

### 3.3 Current Limitations

Current amortized fairness metrics have two primary limitations:

*Insufficient measures of distributional differences between cumulated attention and relevance.* Current definitions compare *expected (average) attention* ($\sum_{t \in \mathcal{T}} a_i^t$) to *expected (average) relevance* ($\sum_{t \in \mathcal{T}} r_i^t$) across rankings, which leads to less reliably fair solutions for attention and relevance distributions where first moments (means) are not sufficient statistics (see Appendix 3).

*Failure to capture the impact of query polarity:* All fairness definitions in the literature currently assume that all attention is good. However, increased attention in the context of queries with negative connotations relative to other similarly relevant individuals can lead to unfairness (see Appendix ??). Hence, incorporating query polarity is necessary to model the real-world impact of unfair rankings.

## 3.4 Problem Statement

We consider (un)fairness to be a function:

$$f: \mathcal{P}(\mathcal{X}) \times \mathcal{P}(\mathcal{Y}) \times \mathbb{R}^T \longmapsto \mathbb{R} \qquad (1)$$

where $A \in \mathcal{P}(\mathcal{X})$ denotes a distribution of cumulative attention and $R \in \mathcal{P}(\mathcal{Y})$ denotes a distribution of cumulative relevance for an individual. *polarity* is a vector containing the polarity of each of $T$ queries over which attention and relevance are cumulated. A lower value is desired.

Our task is to find a class of such functions such that:

$$f(A, R, \text{polarity}) = 0 \implies \text{set of rankings is fair} \qquad (2)$$

We define $f$ to take the form of a scoring function for distribution-aware fairness in ranking (DistFaiR) and identify compatible measures for cumulative attention and relevance distributions in Section 4. We then show that these measures can be modified to depend on query polarity in measurement (Section 5). Lastly, we test the sensitivity of current fair ranking metrics to various query properties such as polarity. This is an important step to assess *fairwashing* effects in rankings [3]. That is, whether fairness measured using query polarity is higher than that measured using query polarity.

## 4 Distribution-aware Fairness in Ranking (DistFaiR)

We propose new distribution-based definitions of amortized fairness. We denote $A_i$ and $R_i$ to be the distribution of an individual $i$'s cumulative attention and relevance till time $T$ respectively. This is in contrast with prior definitions [8, 40, 49], where only the mean of the attention distribution over queries is considered for individuals and groups. We start by defining a class of amortized individual and group unfairness (DistFaiR) and then theoretically characterize a relationship between the two for a class of discrepancy measures.

### 4.1 Defining Amortized Fairness

**DEFINITION 4.1 (DISTFAIR-DIVERGENCE).** *Given two probability distributions $P$ and $Q$ over a common sample space $\Omega$, a divergence $D(P\|Q)$ is a function with the following properties:*

*(1)* Non-negativity: $D(P\|Q) \geq 0$

*(2)* Positivity: $D(P\|Q) = 0$ *if and only if* $P = Q$

**LEMMA 4.2.** *Define the following:*

$$D_{L_1}(P\|Q) = |\mu_P - \mu_Q|$$

*$D_{L_1}$ satisfies definition 4.1 for $P$ and $Q$ when $\mu_P$ and $\mu_Q$ are sufficient statistics for their respective distributions. Additionally, it is jointly convex, subadditive, positively homogeneous, and scales under averages.*

### 4.1.1 Individual Fairness

**DEFINITION 4.3 (AMORTIZED INDIVIDUAL UNFAIRNESS).** *Amortized Individual Unfairness for a set of individuals is defined as the maximum distance between the distributions of cumulative relevance and cumulative attention over a sequence of queries up to time $T$. Specifically, the unfairness is given by:*

$$\text{Unfairness} = \max_{i \in \{1,2,\dots,n\}} D(A_i, R_i),$$

*where $i$ indexes the individuals to be ranked, and $D$ is a divergence.*

Notably, this definition differs from past definitions of amortized fairness [8, 34, 44] as follows: (1) the distribution-based fairness definition allows for distributions attention and relevance that are not fully specified by their means, (2) considers a worst-case notion of individual unfairness. For example, in [8], unfairness is defined to be the $L_1$ distance of difference between cumulative relevance and cumulative exposure scores allocated to a set of $n$ individuals over $T$ queries. In our framework, this is equivalent to choosing a metric $d(P, Q) = |\mathbb{E}[P] - \mathbb{E}[Q]||$, or the absolute difference in expectations of the two distributions. However, this only captures discrepancies between distributions $P, Q$ where means are sufficient statistics, e.g., Guassians with fixed variances or Exponential with rate parameters reciprocal to the mean. Appendix B demonstrates that divergences, which capture properties of distributional difference beyond means, give a more robust and realistic definition of unfairness.

### 4.1.2 Group Fairness

We extend the previous definition to group level by defining the relevance and attention of a group as the average relevance and attention of individuals belonging to that group, respectively. The attention and relevance of a group $g_k \subset [n]$ at time $t$ respectively are random variables:

$$X_{g_k}^t = \frac{1}{|g_k|} \sum_{i \in g_k} X_i^t \quad \text{and} \quad Y_{g_k}^t = \frac{1}{|g_k|} \sum_{i \in g_k} Y_i^t, \qquad (3)$$

where $|g_k|$ denotes the number of individuals in group $g_k$. We can also apply Theorem 3.1 to quantify the tail probability of group level relevance and attention.

The relevance distribution and attention in a group $g_k \subset [n]$ throughout time $t \in \mathcal{T}$ are respectively:

$$X_{g_k} = \sum_{t \in \mathcal{T}} X_{g_k}^t \quad \text{and} \quad Y_{g_k} = \sum_{t \in \mathcal{T}} Y_{g_k}^t. \qquad (4)$$

Denote $A_{g_k}$ and $R_{g_k}$ as the distributions of cumulative attention and relevance from which $X_{g_k}$ and $Y_{g_k}$ are generated.

**DEFINITION 4.4 (AMORTIZED GROUP UNFAIRNESS).** *Amortized Group Unfairness for a set of $G$ groups is defined as the maximum distance between the distributions of cumulative relevance and cumulative attention scores across a sequence of queries up to time $T$ for each group. Each individual is assumed to belong to exactly one of the $G$ groups. Formally, group unfairness is expressed as:*

$$\text{Group Unfairness} = \max_{g_k \in \mathcal{G}} D(A_{g_k} \| R_{g_k}),$$

*where $D$ represents a divergence, $g_k$ denotes the $k$-th group, and $A_{g_k}$ and $R_{g_k}$ represent the distributions of cumulative attention and cumulative relevance for group $g_k$, respectively.*

We refer our definitions of amortized individual and group unfairness above as *DistFaiR*.

## 4.2 Individual Fairness v.s. Group Fairness

THEOREM 4.5. *For any jointly convex DistFaiR divergence that is subadditive, positively homogeneous, and scales under averages, amortized group fairness is upper-bounded by amortized individual fairness. Specifically, we have the following inequality:*

$$\max_{g_k \in \mathcal{G}} D(A_{g_k} \| R_{g_k}) \leq \max_{i \in \mathcal{D}} D(A_i \| R_i) \qquad \forall g_k \in \mathcal{G} \qquad (5)$$

Proof provided in Appendix E.3.

Theorem 4.5 shows that improving individual fairness does not adversely affect group fairness for a sub-class of divergence measures — optimizing for individual fairness can improve group fairness. Although individual fairness is good criteria, it may not always be possible to ensure individual fairness for some divergence measures (e.g., due to computational infeasibility). This also indicates that group fairness constraints could be considered weaker versions of individual fairness, and could be be used more broadly.

## 4.3 Amortized Fairness Re-ranking with Quality Constraints

Theorem 4.5 motivates optimizing for individual (un)fairness. Accordingly, we design an objective function corresponding to individual unfairness to be minimized, similar to Biega *et al.* [8].

$$\min_{M_{i,j}^t} \quad \max_{i \in n} \quad D(A_i \| R_i) \quad \text{(individual fairness)} \qquad (6)$$

$$\text{s.t.} \quad \sum_{j=1}^{k} \sum_{i=1}^{n} \frac{2^{r_i^t} - 1}{\log_2(j+1)} M_{i,j}^t \geq \theta * \rho(t) \quad \text{for each } t \in \mathcal{T} \qquad (7)$$

$$M_{i,j}^t \in \{0, 1\} \quad \forall i, j \qquad (8)$$

$$\sum_i M_{i,j}^t = 1 \quad \forall j \qquad (9)$$

$$\sum_j M_{i,j}^t = 1 \quad \forall i \qquad (10)$$

where $A_i$ and $R_i$ denote cumulative attention and relevance for individual $i$ till time $t$, $M_{i,j}^t$ is a binary variable indicating if individual $i$ is present at rank $j$ for the query at time $t$. $\rho(t)$ indicates the DCG (quality) of the ranking at time $t$. Constraint (5) ensures that the quality of the updated ranking does not decrease beyond a given threshold $\theta$. Additionally, constraints (7) and (8) ensure that each individual can be ranked only once in a ranking and no positions are empty, respectively. Given the large size of the variable space, when $n$ is large, we pre-filter the rankings and set $M_{i,j}^t$ to be fixed when $j > K$ for some known $K \in< n$. Thus, we only re-order the top-$K$ within each ranking.

**Integer Linear Programming Formulation** We solve the above optimization problem using integer linear programming and/or integer quadratic programming. We rely on an open-source toolkit, Gurobi [1] to perform all optimizations where minimizing our objective yields amortized fairness. We study online optimization where a new query arrives at each time $t$, and hence $M_{i,j}^t$ is optimized at each time step, with knowledge of prior assignments [8]. Further details can be found in Appendix F.

Table 1: Summary statistics of all datasets. The relevance score in the `rateMDs` dataset and the query utility score in the `FairTREC2021` datasets are generated using pre-trained LLMs.

| Dataset | #Individuals | #Queries | #Attr. | Relevance | Polarity |
|---|---|---|---|---|---|
| synth-binary | 200 | 15 | 2 | Binary | $\{-1, 1\}$ |
| synth-cont | 200 | 15 | 2 | Cont. | $\{-1, 1\}$ |
| rateMDs | 6.2k | 60 | 2 | Cont. | $\{-1, 1\}$ |
| FairTREC 2021 | 13.5k | 49 | 5 | Binary | $[-1, 1]$ |

## 5 Accounting for Query Polarity

Prior work in fair ranking assumes that all attention is positive [46] and query independent, implying that achieving a higher rank is universally desirable. However, individuals should not be given higher attention for queries with negative connotations than those with similar relevance. Consequently, we extend our fairness definition to account for query properties such as *sentiment polarity* by introducing a *context function* associated with each query.

In this work, we focus on the scalar sentiment polarity associated with each query. Alternative properties may include the clarity of the query, the perceived economic value associated with being highly ranked for the query, etc. This variable will be influenced — at least partially — by the information contained in the query, and may be positive or negative, determining if a higher or lower ranking is more favorable. We also show how this context function can be extended beyond scalar outputs to include a vector of query properties.

Let $\widetilde{X}_i^t$ represent a random variable denoting attention allocated to individual $i$ at time t that incorporates query polarity. Assuming that polarity is searcher-independent (no personalization), it can be decomposed into: (1) the real-world value associated with the attention allocated in response to a query at time $t$ and (2) individual attention. Similar to previous works on fair ranking, we may assume that searcher attention can be modeled well with models like position bias [11].

We denote the context function $\eta(q_t)$ as the polarity associated with query at time $t$. Query polarity-aware attention and relevance allocated to individual $i$ at time $t$ is then:

$$\widetilde{X}_i^t = X_i^t \cdot \eta(q_t) \qquad \text{and} \qquad \widetilde{Y}_i^t = Y_i^t \times \eta(q_t), \qquad (11)$$

where each corresponds to a cumulative distribution $\widetilde{A}_i$ and $\widetilde{R}_i$, respectively. This formulation is free of two assumptions inherent to the exposure-based fairness metrics: (1) the contribution of exposure to amortized ranking is now dependent on properties of the query and (2) exposure can be any real-valued number. Notably, $\eta(q_t) \in \mathbb{R}$, including *negative* values and *zero*, unlike previous work. Then, amortized fairness under DistFaiR can be computed over time, with all notations following from the previous section. We refer to fairness measures defined in the prior section as *query polarity-agnostic*, and those relying on $\eta(t)$ as *query polarity-aware*.

THEOREM 5.1. *Let $X_i^t \sim Bernoulli(p_i^t)$ and $\eta(q_t) \in [a_t, b_t]$; $a_t, b_t \in \mathbb{R}$. With a slight abuse of notation, let $\widetilde{X}_i^t = X_i^t \cdot \eta(q_t) \in [a_t, b_t]$ and*

$$\widetilde{X}_i = \sum_{t \in \mathcal{T}} \widetilde{X}_i^t,$$

*Then, for any $\delta > 0$, we have the following:*

$$P\left(|\widetilde{X}_i - \mathbb{E}[\widetilde{X}_i]| \geq \delta\right) \leq 2\exp\left(-\frac{2\delta^2}{\sum_{t \in \mathcal{T}}(b_t - a_t)^2}\right).$$

REMARK 5.2. *Unlike in Theorem 3.1, the bounds in Theorem 5.1 now depend on both the expected value of the cumulative attention (and relevance) as well as the number of queries observed.*

Thereom 3.1 bounds the likelihood of observing a given deviation $\delta$ from the true polarity-aware cumulative attention for an individual over time $t$. We can apply the same exact bound for polarity-aware cumulative relevance.

## 6 Experiments: Online Fair Ranking

Our experiments are focused on an *online fair ranking setup*, similar to [8]. We assume a realistic setup where a new query arrives at each time $t$, and we re-rank the system-produced ranking at time $t$ to improve fairness. We assume knowledge of attention allocated to individuals in rankings till time $t$ to produce this new fair ranking (i.e., a running memory of cumulative attention per individual).

### 6.1 Experimental Setup

**Datasets** We utilize two synthetic datasets which represent the setting described in the example shown in Figure 4 where female individuals are allocated attention four out of eight rankings, all with negative polarity and two real-world fair ranking datasets [20, 51]; a summary is provided in Table 1 and further details are provided in Appendix D. Our empirical study focuses on post-processing fairness interventions, where individual relevance – or "groundtruth" – scores are known [28].

**Query Properties** We experiment with polarity as the query property. The polarity score is synthetically generated for synth-binary and synth-cont and manually annotated for rateMDs. For the FairTREC 2021 dataset, a pre-trained sentiment classification model is used to generate polarity [6] (see Appendix D).

**Distance Functions** We consider three (pseudo) divergences metrics for measuring unfairness under DistFaiR:

- **$L_1$** distance is defined as the difference between the mean of two distributions: $D_{L_1}(A\|R) = |\mathbb{E}_{X \sim A}[X] - \mathbb{E}_{Y \sim R}[Y]|$.
  - This distance function has been studied in [8], where fairness is computed as the sum of distance values across individuals and is referred to as the inequity of amortized attention (IAA). We note that this function is generally not a proper divergence. However, for distributions $A$ and $R$ whose first moments are sufficient statistics, $D_{L_1}$ satisfies definition 4.1.
- **$L_2^{\text{var}}$** distance is defined as the difference in mean and variance of two distributions:

$$D_{L_2^{\text{var}}}(A\|R) = (\mathbb{E}_{X \sim A}[X] - \mathbb{E}_{Y \sim R}[Y])^2$$
$$+ (\sigma_{X \sim A}[X] - \sigma_{Y \sim R}[Y])^2.$$

We note that $D_{L_2^{\text{var}}}$ benefits from $W_2$, a proper divergence, for two Gaussians, which has the properties for Theorem 4.5.

- **$W_1$** distance is defined as the Wasserstein distance between the distribution of expected attention ($\{a_i^t\}_{t=1}^{\mathcal{T}}$) and distribution of expected relevance ($\{r_i^t\}_{t=1}^{\mathcal{T}}$) for an individual. $D_{W_1}(A\|R) = \frac{1}{T}\sum_{k=1}^{T}|a_i^{(k)} - r_i^{(k)}|$, where $(k)$ denotes the $k$th order statistic of empirical measures $\widehat{A}_i$ and $\widehat{R}_i$ from which each $a_i^t$ and $r_i^t$ is sampled.

### 6.2 Evaluation

We utilized the following fairness criteria.

**Individual Unfairness:** We use three different distance measures defined in Section 6.1 to measure the unfairness as: DistFaiR($L_1$) (IAA), DistFaiR($L_2^{\text{var}}$), and DistFaiR($W_1$). The amortized fairness defined by DistFaiR($L_1$) is similar to the fairness measure studied by [8]. However, we consider the *worst-case* distance between attention and relevance distributions, while [8] consider the average difference across all individuals, which may hide heightened unfairness in some individuals. Our work also generalizes amortized fairness to include appropriate measurements of discrepancies between distributions that require higher-order moments to be specified, i.e., with $L_2^{\text{var}}$ and $W_1$ distances.

**Group Unfairness:** In addition to the group unfairness metrics directly induced by the three distance metrics using Definition 4.4, we consider a standard exposure-based group unfairness definitions: Exposed Utility Ratio (EUR). [40, 49] define the EUR difference as the absolute difference in the ratios of average exposure and average relevance between groups. We also measure an attention parity metric: Demographic Parity[40] (DP).

**Performance** We measure the ranking quality via the DCG@K score, which is the sum of the relevance of the top-K individuals, with a logarithmic discount based on their position:

$$\sum_{k=1}^{K} \frac{r_{\text{rank}(k)}^t}{log_2(k+1)},$$

where rank($k$) returns the index of the individual at rank $k$. After re-ranking, the DCG@K is normalized by the DCG@K of the previous (ideal) ranking to produce a normalized DCG@K between 0 and 1.

### 6.3 Baselines: Fair Re-ranking

We compare the re-ranking performance under DistFaiR metrics to the following baselines: Fairness of Exposure (FoE) [49], Inequity of Amortized Attention (IAA) [8], and Ranking for Individual and Group Fairness Simultaneously (FIGR) [27]. In all cases, the percentage change relative to the original **unconstrained** relevance-ordered rankings are reported (where a positive sign in change indicates a reduction in unfairness post-re-ranking). We perform online optimization in each case, except in FIGR where optimization is on a per query basis (i.e., not amortized).

**IAA**: The method to reduce inequity of amortized attention (IAA) was introduced by Biega *et al.*[8]. An ILP is solved to reduce the absolute difference in the mean of the cumulative attention and cumulative relevance distributions, summed across all individuals. In contrast, our method focuses on *worst-case* minimization.

**Table 2: Individual fairness improves with DistFaiR re-ranking intervention, but the difference depends on the divergence measure used. We show *relative improvement* in fairness post- fair ranking intervention with respect to the original ranking. The columns (i.e., Δ measure) correspond to different fairness measures, while each row corresponds to a fair re-ranking method. We find that post-processing the rankings with DistFaiR improves distribution-based ranking fairness across datasets. Group fairness also improves with DistFaiR in most cases. FoE did not produce an optimal solution on `FairTREC2021`, and hence is not reported. Arrows indicate direction of better performance, with best performance bolded for each fairness metric. *Note that the criterion of the fairness scores varies across cross-columns, so cross-column comparisons are incorrect.***

| Dataset | Method | Relative Change in Individual Fairness (↑) | | | Relative Change in Group Fairness (↑) | | |
|---|---|---|---|---|---|---|---|
| | | $\Delta$ DistFaiR $(L_1)$ | $\Delta$ DistFaiR $(L_1^{var})$ | $\Delta$ DistFaiR $(W_1)$ | $\Delta$ DistFaiR $(L_1)$ | $\Delta$ DistFaiR $(L_1^{var})$ | $\Delta$ DistFaiR $(W_1)$ |
| synth-binary | IAA | **82.50%** | **90.89%** | **68.18%** | 19.75% | 12.58% | 0.80% |
| | FoE | 9.17% | 14.65% | 7.58% | 20.81% | 16.83% | 1.85% |
| | FIGR | 0.00% | 0.00% | 0.00% | 0.00% | 0.00% | 0.00% |
| | DistFaiR($L_1$) | 82.50% | 90.89% | 68.18% | 35.94% | 39.97% | 3.58% |
| | DistFaiR($L_2^{var}$) | 76.38% | **90.89%** | 65.02% | 16.57% | **56.26%** | **7.41%** |
| | DistFaiR($W_1$) | 77.07% | 90.70% | **68.18%** | **66.17%** | 51.47% | 5.02% |
| synth-cont | IAA | 61.75% | **65.12%** | 39.05% | -4.16% | 36.38% | 34.66% |
| | FoE | 2.92% | 5.36% | 0.24% | -72.97% | -64.87% | 1.71% |
| | FIGR | 0.00% | 0.00% | 0.00% | 0.00% | 0.00% | 0.00% |
| | DistFaiR($L_1$) | **61.83%** | 62.41% | 38.34% | 11.00% | **73.88%** | 65.97% |
| | DistFaiR($L_2^{var}$) | 56.93% | 63.81% | 35.67% | **16.59%** | 66.77% | 57.78% |
| | DistFaiR($W_1$) | 50.23% | 59.74% | **39.39%** | -38.68% | 43.62% | 58.88% |
| FairTREC2021 | IAA | 0.00% | 0.00% | -1.19% | **29.06%** | **75.62%** | 10.39% |
| | FIGR | -1.29% | -29.44% | -1.29% | -164.50% | -405.80% | -1.29% |
| | DistFaiR($L_1$) | **0.15%** | **1.27%** | -1.19% | 0.88% | 12.70% | 11.33% |
| | DistFaiR($L_2^{var}$) | **0.15%** | **1.27%** | -0.58% | 18.33% | 66.23% | 2.46% |
| | DistFaiR($W_1$) | **0.15%** | -16.11% | **0.15%** | 25.05% | 73.92% | **16.93%** |
| rateMDs | IAA | 46.60% | 48.73% | -2.36% | 50.00% | 73.30% | 12.77% |
| | FoE | 15.39% | 19.40% | 16.44% | 10.68% | 19.62% | -18.29% |
| | FIGR | -50.07% | -245.38% | -41.29% | -3.93% | -44.50% | -14.99% |
| | DistFaiR($L_1$) | **60.96%** | 69.39% | -0.48% | 41.74% | 64.99% | 21.96% |
| | DistFaiR($L_2^{var}$) | 54.15% | **71.05%** | -3.97% | 61.64% | 79.38% | **26.35%** |
| | DistFaiR($W_1$) | 36.01% | 24.89% | **35.80%** | 64.38% | **86.60%** | 24.40% |

**FoE**: [49] solve a linear program and sample ranking assignments with Birkhoff Von Neumann decomposition [36] to ensure fairness of exposure (FoE). In this algorithm, the quality of rankings is maximized, with the constraint that cumulative attention to relevance ratio is the same for all individuals. We prefilter and re-rank only top-k individuals in each ranking.

**FIGR** [27]: This method jointly aims to reduce "underranking" (which is closely related to individual fairness) in rankings that are post-processed with group fairness constraints. Unlike the other baselines, this is a proportion-based re-ranker, does not explicitly consider attention distributions, and only considers binary groups.

### 6.4 Hyperparameter Tuning

We stratified all datasets into two subsets: 50% tuning and 50% test sets, so no individuals are present in both splits. We re-run each method across two (`fairTREC2021`) or three (all other datasets) such random splits and the average results. All parameters (e.g., $\theta$) are tuned using the tuning split and tested on the test split. We run all optimization algorithms on a 3.2 GHz CPU with 16 GB RAM for $\leq$ 60 minutes. We set K=10 while measuring ranking quality and assume logarithmic discounts in attention till K=10 and zero otherwise. We also only re-rank and optimize for maximum divergence among the top k [8] (here, either 50 or 500) individuals in each ranking. In the online ranking setting, this means that even when the maximum divergence measure across all individuals cannot be reduced by only re-ranking the top-k individuals for a given ranking, we still re-rank to reduce the next possible highest divergence value.

Our experimental flow is as follows: first, we implement our fair ranking definitions (DistFaiR) and compare to baselines. Second, we test if fairness metrics are affected by query polarity. Third, we perform several ablations for, e.g., the fairwashing effect.

## 7 Results

We measure the percentage change in unfairness pre- and post-re-ranking. A positive change – decrease in unfairness – is desired.

**DistFaiR Improves Worst-Case Fairness** Table 2 shows that our re-rankings reduce individual unfairness, when unfairness is measured as the worst-case divergence measure between the attention and relevance distributions across individuals. On all datasets, DistFaiR outperforms or performs on par with IAA. FIGR – which solves a different notion of "underranking" per individual – worsens performance as measured by our metrics. Further, as expected, optimizing the divergence measure itself leads to highest decrease in unfairness (for example, DistFaiR($W_1$) has highest improvement in fairness for the $\Delta$ DistFaiR($W_1$) individual fairness measurement.

Additionally, as seen in Table 3 DistFaiR underperforms IAA-based re-ranking on the IAA metric. This makes sense because DistFaiR focuses on reducing worst-case divergence, while IAA focuses on the average across individuals. Thus, there appear to be tradeoffs between average and worst-case performance. Such

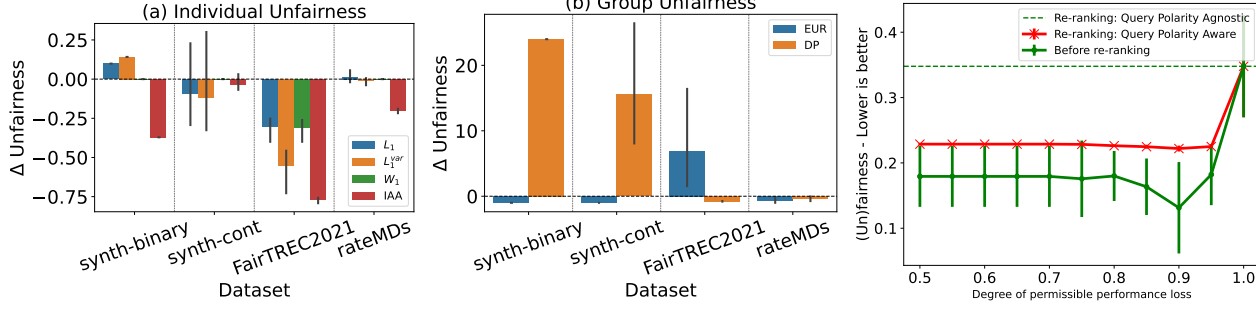

Figure 2: (a) and (b) show the difference between fairness metrics measured with and without query polarity. Query polarity impacts all amortized fairness metrics, as they differ from zero as seen in the plots. (rightmost) We plot the re-ranking performance of polarity agnostic and aware re-rankings under different permissible performance loss changes for the synth-cont dataset (DistFaiR($L_1$)), where we can see polarity agnostic re-ranking underperforms polarity aware re-ranking.

observations have also been made in other fairness contexts [58]. Lastly, from Table 3, DistFaiR improves or retains EUR.

**Distance Function is an Important Design Choice.** Our results show that the distance is an important design choice. We find that the performance of $L_1$ and $L_2^{\mathrm{var}}$ are close (e.g., synth-binary). We hypothesize that the optimization with $W_1$ is more difficult, due to which performance improvements are smaller. Note that $L_2^{\mathrm{var}}$ is the $W_2$ solution under assumptions of gaussianity. Hence, it is possible that using the $L_2^{\mathrm{var}}$ measure could be an easier objective, but we can remove the distribution assumption for the general $W_2$.

**Individual Fairness Not at Odds with Group Fairness.** We also find that reducing individual unfairness under DistFaiR leads to reductions in group unfairness in most cases (Table 2). While group unfairness does increase in some cases, the degree of change cannot exceed a specific limit (upto individual unfairness) as per our theoretical findings. We also see similar trends on standard group fairness of EUR [40] (see Table 3 in Appendix) for three datasets.

**Online vs Offline Optimization.** In Figure 6(c) in the Appendix, we observe that fully offline optimization reduces unfairness in a batch of rankings more effectively than fully online. This is meaningful because even if the full set of queries may not be known apriori, a batch of likely queries may be known. Additionally, variance in online fairness is lower when optimizing for divergence including higher order moments (e.g. $W_1$; see Appendix Figure 7).

**Fairness Metrics are Sensitive to Query Polarity.** In Figure 2 (a) and (b), we compute the difference between fairness metrics measured with and without query polarity. When the difference is positive, this indicates fairwashing, where rankings seem more fair than they actually are. In all cases, we compute the percentage in change. We observe that all fairness metrics, for both individual and group fairness, are sensitive to query polarity. If one relies on the query polarity agnostic metrics, conclusions regarding the unfairness of the rankings would be incorrect. That is, fairwashing may occur.

**Ranking Quality and Fairness Tradeoff.** We plot the variation in fairness across thresholds of allowable ranking quality loss ($\theta$). Lower unfairness is observed at lower $\theta$ for the polarity-aware re-ranking (Figure 2 (c)), indicating a ranking quality and fairness

tradeoff. Additionally, we plot the polarity agnostic re-ranking performance in Figure 2(c), which leads to higher (worse) unfairness than using query polarity. This matches our discussion that fairness metrics are sensitive to query polarity, and polarity agnostic re-ranking may harm the actual (un)fairness. Higher standard deviation is observed in polarity-aware re-ranking.

Importantly, in many real-world applications, different queries may have multiple differing real-world properties beyond polarity. Accordingly, we generalize our distribution-aware fairness definition to allow multiple query properties as a vector, where multiple queries form a high-dimensional distribution. Initial results with this setup for the synthetic datasets are in the Appendix K.

## 8 Conclusions

In this paper, we propose a new distribution-aware distance-based metric, DistFaiR, for amortized fairness measurement. We identify metrics under DistFaiR with the useful property that group and individual fairness are not at odds. Accordingly, we propose an integer linear programming-based re-ranking to improve fairness based on prior work by [8] while maintaining similar ranking quality. We find optimizing our objective improves both group and individual fairness. We also highlight query properties that have been ignored so far in fair-ranking literature, where not considering these properties can lead to fairwashing. We empirically demonstrate fairwashing effects due to a lack of query polarity consideration and propose/evaluate a method to mitigate this effect. Future work includes a formulation of a fully differentiable approach.

We make normative assumptions that the distribution of attention should be close to that of relevance. However, a different link function may be more appropriate [46]. Additionally, scores allotted to minority groups may be under-estimates of their true value [35, 43] and may need to be pre-processed [37]. Importantly, there may not be purely technical fixes for operationalizing real-world fair ranking [22]. Our approach, we believe, is a step towards reducing the scale of such issues.

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

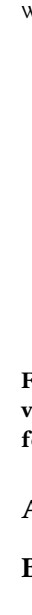
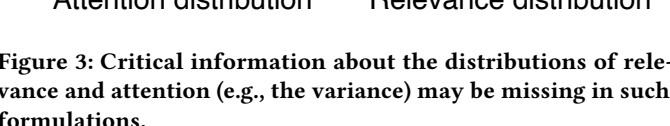

**Figure 3: Critical information about the distributions of relevance and attention (e.g., the variance) may be missing in such formulations.**

## A  Overview Figure

## B  Information Loss with Expectation-only Approaches

We observe that critical information about distributions of attention may be missing from prior formulations e.g., as shown in Figure 3, where the mean attention may match the mean relevance scores for an individual, but the distributions may be very dissimilar. Consider two distributions $A$ and $R$ defined as follows

$$A = \mathcal{N}(0, \Sigma) \qquad \text{and} \qquad R = 0.5\mathcal{N}(-\mu, \Sigma) + 0.5\mathcal{N}(\mu, \Sigma).$$

Also, define

$$\widetilde{R} = \mathcal{N}(0, \Sigma)$$

Clearly $\mu_A - \mu_R = \mu_A - \mu_{\widetilde{R}} = 0$. However, the distribution between attention and relevance is clearly not the same. Particularly, suppose $\mu = 3$. Then $\text{Var}(A) < \text{Var}(R)$, and attention is spread out much less broadly across individuals as relevance. Thus, attention is much more concentrated for some individuals, while relevance is not concentrated within the same individuals. Importantly, fairness metrics that only consider means would consider this setting fair.

Let $\Sigma = I$ and

$$\Gamma(x) = \left( \frac{\exp\left(-\frac{x^2}{2}\right)}{0.5\exp\left(-\frac{(x+\mu)^2}{2}\right) + 0.5\exp\left(-\frac{(x-\mu)^2}{2}\right)} \right).$$

Then

$$D_{\text{KL}}(A\|R) = \int_{-\infty}^{\infty} \frac{1}{\sqrt{2\pi}} \exp\left(-\frac{x^2}{2}\right) \log \Gamma(x) dx. \tag{12}$$

Clearly, $D_{\text{KL}}(A\|\widetilde{R}) = 0 < D_{\text{KL}}(A\|R)$, better measuring the discrepancy between cumulative attention and relevance, unlike mean distance measures in previous work.

## C  Example of Fairwashing

As shown in Figure 4, past formulations of amortized fair ranking may be prone to fairwashing.

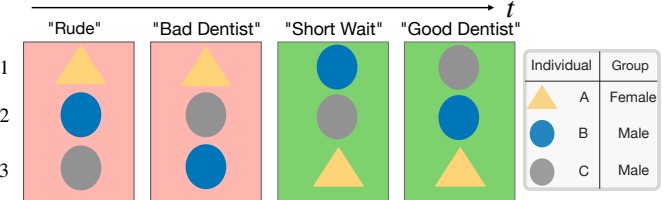

**Figure 4: Past work in amortized fair ranking has ignored the impact of query polarity. Here, if all individuals are equally relevant, and expected attention scores for ranks 1,2,3 are $\{0.5, 0.5, 0\}$ respectively, the sequence of queries appear fair because an individual's expected attention accumulated over the four queries is proportional to their relevance. However, we observe that the female doctor is allocated attention only for the queries with negative polarity ("rude", "bad dentist"). This leads to _fairwashing_.**

## D  Datasets

We utilize four datasets used in our experiments. In each dataset, relevance scores are normalized to form a distribution within a ranking. `fairtrec2021` is licensed under the CC BY-SA 3.0 license. `rateMDs` is released as a part of open-sourced research publication [51].

### D.1  Synthetic Hiring Datasets

Two synthetic datasets are generated, to mimic the example shown in Figure 4. The sensitive attribute is this dataset is the sex of the individual being ranked. We generate two versions, one with binary relevance scores, and one where in one case the relevance is continuous. In the binary relevance dataset (`synth-hiring1`) the relevance for individuals is either 1.01 or 0.99, with male individuals having a relevance score of 1.01 for queries with positive polarity, and 0.99 for queries with negative polarity. In the continuous dataset (`synth-hiring2`), the relevance in each group is sampled from a uniform distribution, with range $[1, 1.1]$ for the less priortized group, and $[1, 1.2]$ for the more prioritized group per query. Each query in the dataset has a polarity value $\in \{-1, 1\}$.

### D.2  RateMDs

We also utilize a healthcare dataset [51] for ranking doctors corresponding to a text query. The sensitive attribute is sex. Each ranking corresponds to a text query such as "best dentist". The ranking is produced by using a pre-trained LLM model[1] to match the text to reviews of doctors, and order the doctors in decreasing order of ranking score. Each query is associated with a utility $\in \{-1, 1\}$, which is annotated based on the sentiment polarity of the query (positive sentiment: 1, negative sentiment: -1). Note that some queries were specific to a speciality: e.g., "best dentist", and relevance scores were produced for all doctors in the dataset (i.e., including doctors who have a different specialty). Thus, our results produced are highly influenced by the correctness of the LLM-driven scores. Ideally, the scores allocated to doctors from different

---

[1]https://huggingface.co/cross-encoder/ms-marco-TinyBERT-L-2

specialties will be low. We leave experiments with varying how the relevance score is produced to future work.

## D.3 FairTREC2021

In this dataset, the items correspond to Wikipedia articles, and the query is the corresponding article domain [20]. This is a standard dataset used in multi-query ranking tasks. The sensitive attribute is geographic location(s) referred to in the data, which we categorize into one of five groups. The query utility score is a continuous score $\in [-1, 1]$, and is produced by a pre-trained sentiment classification model [6]. Specifically, the utility score is the sum of the sentiment polarity of the query – where -1 denotes negative, 0 denotes neutral, and 1 denotes positive – weighted by predicted probabilities of each polarity class.

## E Individual vs Group Fairness

### E.1 Tail Probability Bounds for Cumulative Attention and Relevance

THEOREM E.1. *Let* $X_i^t \sim Bernoulli(p_i^t)$ *and*
$$X_i = \sum_{t \in \mathcal{T}} X_i^t.$$
*The expected value of* $X_i$ *is given by:*
$$\mathbb{E}[X_i] = \sum_{t \in \mathcal{T}} p_i^t.$$
*Then, for any* $\delta > 0$, *we have the following:*
$$\mathbb{P}\left(|X_i - \mathbb{E}[X_i]| \geq \delta\mathbb{E}[X_i]\right) \leq 2\exp\left(-\frac{\delta^2\mathbb{E}[X_i]}{2+\delta}\right).$$

PROOF. Assume that $X_i^t$'s are independent for different $t$ and observe that the domain of random variable $X_i^t$ is $\{0, 1\}$, i.e., bounded and non-negative. Using Chernoff bounds for the sum of independent Bernoulli random variables we have upper and tail bounds, respectively:

$$P(X_i \geq (1+\delta)\mathbb{E}[X_i]) \leq \exp\left(-\frac{\delta^2\mathbb{E}[X_i]}{2+\delta}\right),$$

$$P(X_i \leq (1-\delta)\mathbb{E}[X_i]) \leq \exp\left(-\frac{\delta^2\mathbb{E}[X_i]}{2}\right).$$

Applying a union bound for both the upper and lower tails, we have:

$$P\left(|X_i - \mathbb{E}[X_i]| \geq \delta\mathbb{E}[X_i]\right) \leq 2\exp\left(-\frac{\delta^2\mathbb{E}[X_i]}{2+\delta}\right).$$
□

### E.2 Proof of Lemmas

LEMMA E.2. *Define the following:*
$$D_{L_1}(P\|Q) = |\mu_P - \mu_Q|$$

$D_{L_1}$ *satisfies definition 4.1 for* $P$ *and* $Q$ *when* $\mu_P$ *and* $\mu_Q$ *are sufficient statistics for their respective distributions. Additionally, both are jointly convex.*

PROOF. We prove that $D_{L_1}(P\|Q) = |\mu_P - \mu_Q|$ satisfy the following properties:

**1. Non-negativity:**
For $D_{L_1}(P\|Q)$, the expressions involve absolute values, which are non-negative by definition. Thus,
$$D_{L_1}(P\|Q) = |\mu_P - \mu_Q| \geq 0. \tag{13}$$

**2. Positivity:**
For $D_{L_1}(P\|Q) = |\mu_P - \mu_Q|$, we have $D_{L_1}(P\|Q) = 0$ if and only if $\mu_P = \mu_Q$. Since $\mu_P$ and $\mu_Q$ are sufficient statistics, $\mu_P = \mu_Q$ implies $P = Q$, and conversely, if $P = Q$, then $\mu_P = \mu_Q$.

**3. Joint convexity:**
Let $P_\lambda = \lambda P_1 + (1-\lambda)P_2$ and $Q_\lambda = \lambda Q_1 + (1-\lambda)Q_2$, where $\lambda \in [0, 1]$. The mean is a linear functionals of the distributions, so:
$$\mu_{P_\lambda} = \lambda\mu_{P_1} + (1-\lambda)\mu_{P_2}.$$
For $D_{L_1}(P\|Q) = |\mu_P - \mu_Q|$, we use the convexity of the absolute value function:
$$|\mu_{P_\lambda} - \mu_{Q_\lambda}| \leq \lambda|\mu_{P_1} - \mu_{Q_1}| + (1-\lambda)|\mu_{P_2} - \mu_{Q_2}|.$$
Thus, $D_{L_1}(P_\lambda\|Q_\lambda) \leq \lambda D_{L_1}(P_1\|Q_1) + (1-\lambda)D_{L_1}(P_2\|Q_2)$.

**4. Subadditivity:** For $D_{L_1}(P\|Q)$, we need to verify:
$$D_{L_1}(P\|R) \leq D_{L_1}(P\|Q) + D_{L_1}(Q\|R)$$
This becomes:
$$|\mu_P - \mu_R| \leq |\mu_P - \mu_Q| + |\mu_Q - \mu_R|$$
This is the standard triangle inequality for absolute values, so subadditivity holds.

**5. Scaling over averages:** For $D_{L_1}(P\|Q)$, scaling over averages refers to how the divergence behaves when comparing averages (means) of distributions. It requires:
$$D_{L_1}\left(\frac{P_1 + P_2}{2}\,\Big\|\,\frac{Q_1 + Q_2}{2}\right) \leq \frac{D_{L_1}(P_1\|Q_1) + D_{L_1}(P_2\|Q_2)}{2}$$
This becomes:
$$\left|\frac{\mu_{P_1} + \mu_{P_2}}{2} - \frac{\mu_{Q_1} + \mu_{Q_2}}{2}\right| \leq \frac{|\mu_{P_1} - \mu_{Q_1}| + |\mu_{P_2} - \mu_{Q_2}|}{2}$$
Again, this holds due to the triangle inequality for absolute values.

**6. Positive homogeneity:** For $D_{L_1}(P\|Q)$:
$$D_{L_1}(\alpha P\|\alpha Q) = |\alpha\mu_P - \alpha\mu_Q| = \alpha|\mu_P - \mu_Q| = \alpha D_{L_1}(P\|Q)$$
Thus, positive homogeneity holds.
□

### E.3 Individual Fairness Upper-Bounds Group Fairness

THEOREM E.3. *For any jointly convex DistFaiR divergence that is subadditive, positively homogeneous, and scales under averages, amortized group fairness is upper-bounded by amortized individual fairness. Specifically, we have the following inequality:*
$$\max_{g_k \in \mathcal{G}} D(A_{g_k}\|R_{g_k}) \leq \max_{i \in \mathcal{D}} D(A_i\|R_i) \quad, \tag{14}$$

where $A$ and $R$ are distributions that denote attention and relevance, respectively, individuals $i \in \{1, \ldots, n\}$, and $g_k$ denotes the set of individuals $i$ that belong to group $k$.

PROOF. Let $A_i, R_i$ denote the distributions of random variables $X_i, Y_i$, respectively.

**Assume $D$ is subadditive, positively homogeneous, and scales under averages.**

Denote

$$X_{g_k} = \frac{1}{|g_k|} \sum_{i \in g_k} X_i \quad \text{and} \quad Y_{g_k} = \frac{1}{|g_k|} \sum_{i \in g_k} Y_i, \qquad (15)$$

such that, by scaling property of $D$,

$$X_{g_k} \sim A_{g_k} \quad \text{and} \quad Y_{g_k} \sim R_{g_k}. \qquad (16)$$

Denote

$$X_i' = \frac{1}{|g_k|} X_i \quad \text{and} \quad Y_i' = \frac{1}{|g_k|} Y_i$$

s.t.

$$X_i' \sim A_i' \quad \text{and} \quad Y_i' \sim R_i'.$$

$A_{g_k} = A_1' \circ A_2' \circ \ldots \circ A_{|g_k|}'$ and $R_{g_k}' = R_1' \circ R_2' \circ \ldots \circ R_{|g_k|}'$, where $\circ$ denotes convolution. Recall that $X_i$'s and $Y_i$'s are independent.

$$D(A_{g_k} \| R_{g_k}) \le \sum_{i \in g_k} D(A_i' \| R_i') \qquad (17)$$

$$= \frac{1}{|g_k|} \sum_{i \in g_k} D(A_i \| R_i) \qquad (18)$$

$$\le \max_{i \in g_k} D(A_i \| R_i) \qquad (19)$$

$$\le \max_{i \in \mathcal{D}} D(A_i \| R_i), \qquad (20)$$

where Equation 17 is a result of subadditivity and Equation 18 is a result of positive homogeneity.

Taking the max over all groups,

$$max_{g_k \in \mathcal{G}} D(A_{g_k} \| R_{g_k}) \le \max_{i \in \mathcal{D}} D(A_i \| R_i), \qquad (21)$$

completes the proof.

□

### E.4 Tail Probability Bounds for Polarity-Aware Cumulative Attention and Relevance

THEOREM E.4. *Let $X_i^t \sim Bernoulli(p_i^t)$ and $\eta(q_t) \in [a_t, b_t]$; $a_t, b_t \in \mathbb{R}$. With a slight abuse of notation, let $\widetilde{X}_i^t = X_i^t \cdot \eta(q_t) \in [a_t, b_t]$ and*

$$\widetilde{X}_i = \sum_{t \in \mathcal{T}} \widetilde{X}_i^t,$$

*The expected value of $\widetilde{X}_i$ is given by:*

$$\mathbb{E}[\widetilde{X}_i] = \sum_{t \in \mathcal{T}} \eta(q_t) \cdot p_i^t.$$

*Then, for any $\delta > 0$, we have the following:*

$$P\left( |\widetilde{X}_i - \mathbb{E}[\widetilde{X}_i]| \ge \delta \right) \le 2 \exp\left( -\frac{2\delta^2}{\sum_{t \in \mathcal{T}} (b_t - a_t)^2} \right).$$

PROOF. Assume that $\widetilde{X}_i^t$'s are independent for different $t$ and observe that each $\widetilde{X}_i^t \in [a_t, b_t]$, i.e., bounded. Using Hoeffding's inequality for the sum of independent bounded random variables, we have:

$$P\left( \widetilde{X}_i \ge \mathbb{E}[\widetilde{X}_i] + \delta \right) \le \exp\left( -\frac{2\delta^2}{\sum_{t \in \mathcal{T}} (b_t - a_t)^2} \right),$$

$$P\left( \widetilde{X}_i \le \mathbb{E}[\widetilde{X}_i] - \delta \right) \le \exp\left( -\frac{2\delta^2}{\sum_{t \in \mathcal{T}} (b_t - a_t)^2} \right).$$

By applying a union bound for the upper and lower tails, we get:

$$P\left( |\widetilde{X}_i - \mathbb{E}[\widetilde{X}_i]| \ge \delta \right) \le 2 \exp\left( -\frac{2\delta^2}{\sum_{t \in \mathcal{T}} (b_t - a_t)^2} \right).$$

□

## F Integer Linear Programming Formulation

The solver for each optimization relies on the branch-and-bound linear programming [54] algorithm wherein the linear programming relaxation (i.e., without integrality constraints) is first solved. Then, a tree-based search is performed to find feasible solutions, valid upper bounds (best found objective) and the best possible bound. The gap between the best found objective and best possible bound – referred to as the optimality gap [9] – is measured during each search iteration. The Gurobi solver also uses routines of presolve [2], cutting planes [38], etc. to make the optimization more efficient.

## G Individual Fairness Bounds Group Fairness under DistFaiR.

Theorem 4.5 shows that group (un)fairness is upper-bounded by individual (un)fairness for some classes of distance functions. In Figure 5(a), we experimentally validate this by computing group and individual unfairness for the $W_1$ divergence measure.

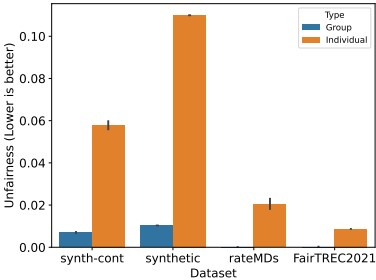

**Figure 5: Individual Fairness Bounds Group Fairness under DistFaiR (here, $DistFaiR(W_1)$)**

## H    Note on Query Polarity Aware Ranking

Note that we can also utilize normalized versions of query utility $U(t)$, e.g., with softmax-based normalization. However, normalization in this manner ensures may convert queries with negative polarity to positive – thus we use unnormalized scores in our experiments.

## I    Additional Results: Impact of Re-ranking on Group Fairness

Re-ranking interventions optimized to improve individual fairness tend to improve or retain EUR as seen in Table 3. Importantly, DistFaiR underperforms IAA on the IAA individual fairness measurement which makes sense because DistFaiR focuses on worst-case distance between individuals, while IAA focuses on average across individuals. Thus, there are tradeoffs between average and worst-case performance as seen in other fairness contexts [58]. Note that we set degree of permissible performance loss (i.e., least possible nDCG) to 80%, which all methods exceed. However, IAA does have higher performance on FairTREC2021.

Table 3: DistFaiR also improve IAA and EUR in a majority of cases (here, positive, higher is better). However, there are some distance function-dependent variations. We show difference in group unfairness, when compared to the unconstrained ranking. IAA outperforms DistFaiR on the IAA fairness measurement

| Dataset | Baseline | Fairness | | nDCG@10 |
|---|---|---|---|---|
| | | IAA | EUR | |
| synth-binary | IAA | **68.88**% | 19.75% | 100% |
| | FoE | 12.89% | 20.81% | 100% |
| | DistFaiR($L_1$) | 57.62% | 35.94% | 100% |
| | DistFaiR($L_2^{gar}$) | 39.30% | 16.57% | 100% |
| | DistFaiR($W_1$) | 45.91% | **66.17**% | 100% |
| synth-cont | IAA | **46.44**% | -4.16% | 91% |
| | FoE | 3.56% | -72.97% | 99% |
| | DistFaiR($L_1$) | 29.62% | 11.00% | 88% |
| | DistFaiR($L_2^{gar}$) | 34.81% | **16.59**% | 88% |
| | DistFaiR($W_1$) | -4.10% | -38.68% | 86% |
| FairTREC2021 | IAA | **0.88**% | 21.96% | **96**% |
| | DistFaiR($L_1$) | -3.49% | 30.66% | 82% |
| | DistFaiR($L_2^{gar}$) | -2.47% | -5.95% | 84% |
| | DistFaiR($W_1$) | -2.36% | 38.23% | 84% |
| rateMDs | IAA | **11.63**% | 50.00% | 91% |
| | FoE | -1.01% | 10.68% | **93**% |
| | DistFaiR($L_1$) | -2.95% | 41.74% | 86% |
| | DistFaiR($L_2^{gar}$) | 1.24% | 61.64% | 87% |
| | DistFaiR($W_1$) | -13.16% | **64.38**% | 83% |

## J    Fairness Over Time

We observe that variance in online fairness is generally lower for divergence measures that use higher order moments as seen in Figure 7.

## K    Multiple properties per query

We empirically conduct experiments where each query in the synthetic dataset contains three total properties. We define fairness as the sum of fairness metrics with each component separately.

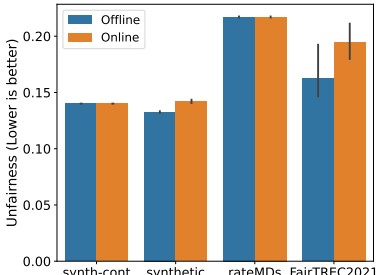

Figure 6: Online fair ranking – where queries arrive one after the other for ranking – underperforms offline fair ranking where the whole set of queries is known apriori. However, the degree of difference is small.

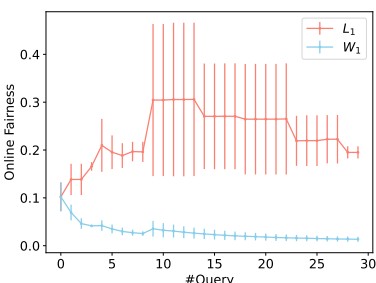

Figure 7: Online fair ranking fairness on rateMDs.

Table 4: Impact of fair ranking

| Dataset | Measure | Pre-intervention | Post-intervention |
|---|---|---|---|
| synth-binary | IAA | 12.80 | 3.98 |
| | DistFaiR($L_1$) | 0.80 | 0.14 |
| | DistFaiR($W_1$) | 0.11 | 0.03 |
| synth-cont | IAA | 7.58 | 3.82 |
| | DistFaiR($L_1$) | 0.40 | 0.14 |
| | DistFaiR($W_1$) | 0.06 | 0.04 |

We observe that online optimization reduces unfairness from across metrics in the synth-binary and synth-cont datasets in Table 4.

## L    Online vs Offline Optimization

We observe that online optimization underperforms fully offline fairness optimization, but only by a small margin.

