# OpenReview forum: "What’s in a Query: Polarity-aware Distribution-based Fair Ranking"
_ACM.org/TheWebConf/2025/Conference — WWW 2025 Poster_

### Official Review · Reviewer_jQBR · 2024-11-11

**Novelty:** 4
**Technical Quality:** 5

**Review:**

### quality
The paper proposes a new ranking method that is dedicated to reducing individual unfairness and thereby reducing unfairness between groups, while also considering the polarity of the search problem itself. In the experiments, two synthetic datasets and two real-world datasets were used for comparison with traditional baseline methods。

### clarity
The paper provides a clear description, explaining the problem to be solved and previous work. It also clearly describes its own processing steps. However, it does not provide the source code, which makes reproduction somewhat difficult.

### originality
This paper proposes a distribution-aware fairness measurement method called DistFaiR, which is different from previous methods that only considered the differences in average attention and correlation. By introducing various divergence measures (such as $D_{L_1} $ , $D_{L_2^{var}}$, and $D_{W_1}$), it characterizes the differences between the distributions of attention and relevance, focusing not only on the mean but also considering the impact of higher-order moments (such as variance, skewness, etc.).The paper incorporates polarity factors into the calculation of attention and relevance by introducing a context function $\eta(q_t)$ associated with the query

### pros
1. The process description is clear, with each step described using mathematical formulas, leaving no room for ambiguity. A comparison with baseline algorithms was also made.
2. Considering the limitations of using only the mean value, multiple metrics have been proposed to further measure fairness. Additionally, the polarity in the search process has been taken into account and applied in the processing.

### cons
1. The paper did not provide the source code, which may lead to difficulties in reproducing the results.
2. The meaning of (a) and (b) in Figure 1 is unclear.
3. Should there be a figure that illustrates the flow of the algorithm to help understand how section 4 and section 5 work together.
4. In section 6.2, the results of performance do not seem to be given.
5. (c) is not marked in Figure 2. And Figure 2 (c) are the results of a synthetic data set; it is better to give more test results or use a public data set.
6. In the experimental part, all the metrics proposed in the paper are used instead of some recognized metrics.

**Questions:**

1. Do the different columns of a in Figure 2 represent different metrics? Which algorithm is used to measure the metrics of a and b?
2. Which  fairness measures are used in Figure 6 and 7?
3. In Equation 11, both attention and relevance are multiplied by the polarity function $\eta(q_t)$. If it is a positive question, will it not affect the ranking? If it is a negative question, will it reverse the ranking?
4. To address the issue of unfairness, should we start with the calculation of relevance? Merely analyzing and changing rankings based on polarity alone does not provide any information about the individuals being ranked. Can this truly improve the phenomenon of unfairness?

**Reviewer Confidence:**

2: The reviewer is willing to defend the evaluation, but it is likely that the reviewer did not understand parts of the paper

**Scope:**

4: The work is relevant to the Web and to the track, and is of broad interest to the community

---

### Official Review · Reviewer_HYmE · 2024-11-24

**Novelty:** 6
**Technical Quality:** 6

**Review:**

### **Pros**:
* The paper proposes a novel approach to measure fairness in automated ranking systems. Unlike existing approaches that only consider the mean of attention and relevance, DistFaiR accounts for high-order moment, such as variance and skewness. This makes the fairness measurements more robust and address limitation in existing methods that only focus on averages.
* Another contribution of the paper is the inclusion of query polarity in the fairness metrics. Many systems assume that all attention is inherently positive, which can lead to fairwashing, where rankings seem fairer than they truly are. The authors address this risk by incorporating the polarity of queries, ensuring that negative attention is not unfairly directed at certain individuals or items.
* The paper provides a solid theoretical foundation, showing that individual fairness serves as upper bound for group fairness for a class of divergence measures under their framework. This result supports the idea that focusing on fairness for each individual can lead to better fairness for entire groups as well.
* The authors validate their approach using both synthetic and real-world datasets. The results show the effectiveness of their approach compared to existing baseline methods.
* The paper addresses a critical issue in high-stakes settings such as hiring, healthcare, online platforms, where fairness in ranking systems has profound societal implications.

### **Cons**:
* Some parts of the paper, especially the explanation of DistFaiR and the role of query polarity, are a bit difficult to understand. More examples or a clear explanation (like a step-by-step example) could help. The way polarity-aware fairness is included in the model could also be made easier to understand with more intuitive visualizations or examples.
* The use of integer linear programming for optimization may not scale well with large datasets. The paper could explore the computational trade-offs of using ILP in large-scale applications.
* The datasets used in the experiments might not show full complexity of real-world scenarios. For example, real-world rankings may have more queries and more complex contexts than just polarity.
* While the paper relies on theory and experiments with existing datasets, user studies could help validate whether the proposed fairness metrics align with human perceptions of fairness in rankings.
* The paper assumes that the distribution of attention should closely match relevance. The authors briefly mention that a different link function might be better in some cases, but this idea needs more discussion.
* Though the paper focuses on query polarity, it does not explore other factors that might affect fairness. This is only briefly mentioned, it could be discussed more.
* The paper assumes that each individual belongs to just one group. However, in real-world, people can belong to multiple groups or have overlapping identities, which could complicate fairness considerations.

**Questions:**

1.  How does the proposed framework handle real-world scenarios where queries involve multiple dimensions, such as polarity, economic value (e.g., related to business or shopping), or sensitivity (personal or sensitive information) ? Since the datasets used in the experiments may not fully capture the complexity of real-world rankings, how well does DistFaiR adapt to more diverse and complex query contexts? While the paper focus on query polarity, are there plans to extend the framework to include other factors that might impact fairness, such as query urgency or user intent?

2. Given that integer linear programming can be computationally intensive for large datasets or high-frequency queries, can you address the trade-offs using ILP in large-scale applications?

3.  How reliable is the polarity annotation process? If a query’s polarity is misclassified (e.g., a positive query labeled as neutral or vice versa), how would this impact the fairness of the ranking and the potential risk of fairwashing?

4. The paper assumes each individual belongs to a single group, but in real-world situations, individuals may belong to multiple overlapping groups or have intersectional identities. How does DistFaiR handle such complexities, and can the framework be adjusted to account for them?

5. The paper assumes that the distribution of attention should closely match relevance. What are potential consequences if this assumption does not hold in some contexts? How flexible is the framework in adapting to cases where this assumption may not be valid?

6. While the paper provides a solid theoretical and empirical foundation, how do you plan to validate whether the fairness metrics used in DistFaiR align with human perceptions of fairness? Can you discuss briefly how user studies could be conducted to gather more subjective feedback on the effectiveness of these fairness measurements?

**Minor issues**:
* In page 3, the sentence just before the last one “However, increased attention in the context of queries with negative connotations relative to other similarly relevant individuals can lead to unfairness (see Appendix ??).”, has an issue with the appendix number, which needs to be corrected.
* In the experimental setup section, I noticed that the sensitive attributes (or groups) considered for each dataset were not mentioned. This information should be included for clarity.

**Reviewer Confidence:**

3: The reviewer is confident but not certain that the evaluation is correct

**Scope:**

4: The work is relevant to the Web and to the track, and is of broad interest to the community

---

### Official Review · Reviewer_hZM3 · 2024-11-25

**Novelty:** 5
**Technical Quality:** 6

**Review:**

The paper presents a re-ranking method, together with a ranking fairness metric DistFaiR. The paper puts the contributions on a solid theoretical foundation and evaluates the presented approaches on multiple datasets. The conclusions that can be drawn are presented in a clear and concise way. It is, however, not entirely clear how the proposed approach could be implemented in practice when queries are not necessarily predictable and the data to be ranked could change over time.

**Questions:**

How would this approach be implemented in practice when queries cannot be easily predicted in advance and the underlying data can change over time? Do the authors have a practical implementation of their method that they could make available to the community?

**Reviewer Confidence:**

2: The reviewer is willing to defend the evaluation, but it is likely that the reviewer did not understand parts of the paper

**Scope:**

3: The work is somewhat relevant to the Web and to the track, and is of narrow interest to a sub-community

---

### Official Review · Reviewer_p3CP · 2024-11-30

**Novelty:** 6
**Technical Quality:** 6

**Review:**

This paper considers amortized fair rankings where relevance and attention are cumulated over a sequence of queries to generate rankings. However, in contrast to the existing methods which consider expected amortized attention, the proposed method considers divergence-based measures operating in attention distribution rather than point estimation measures. The definition also considers both polarities in queries, which makes for a more realistic setting. The authors also prove that group fairness is upper bounded by individual fairness under this definition.

The paper is very well written and makes significant contributions to the fair ranking literature. Incorporating polarity in queries as well as considering distribution rather than point estimates also make their definition more robust than the existing ones. The paper also introduces key theoretical results which are also very interesting.

I do have a couple of points which the authors could consider.

1. If the authors could show some empirical results to demonstrate the superiority of distribution based measures over point estimate based measures, it would further demonstrate the effectiveness of the proposed method.

2. The experiments are carried out on synthetic datasets. The authors could also consider some real world datasets.
Perhaps - https://github.com/MilkaLichtblau/xing_dataset/blob/master/README.md

**Questions:**

I don't have any additional questions to the authors. The paper is already very clear and well written.

**Reviewer Confidence:**

3: The reviewer is confident but not certain that the evaluation is correct

**Scope:**

4: The work is relevant to the Web and to the track, and is of broad interest to the community

---

### Official Review · Reviewer_74e5 · 2024-12-01

**Novelty:** 3
**Technical Quality:** 4

**Review:**

This paper recognizes the missing of critical information about the distributions of attention and properties of queries (such as query polarity) in existing fair ranking literature and propose DistFairR to overcome it.

The writing of this paper is not so good, with many concepts used before defined or described, e.g. what is the formal definition of attention, why can it formulated as a single binary variable?

Regarding assumptions: why do you make assumptions like "We assume that 𝑋𝑡𝑖 and 𝑋𝑡𝑗 are independent ∀𝑡 when 𝑖 ≠ 𝑗. " It is neither explicitly mentioned in [8], nor [32] (which only mentions "Note that assuming a Binomial distribution has to be taken with a grain of salt. It assumes independence between preferences statements, which is not necessarily the case for statements derived from the same query."). But you mention such an assumption is similar. Why similar?

**Questions:**

Refer to the review.

**Reviewer Confidence:**

4: The reviewer is certain that the evaluation is correct and very familiar with the relevant literature

**Scope:**

3: The work is somewhat relevant to the Web and to the track, and is of narrow interest to a sub-community